# Differential Post-Fire Recovery of Tree and Shrub Growth and Water-Use Efficiency in a Mediterranean Coastal Dune System

Jesús Julio Camarero [1,*] , Ricardo Díaz-Delgado [2] , Michele Colangelo [1,3] , Cristina Valeriano [1] , Raúl Sánchez-Salguero [1,4] and Javier Madrigal [5,6]

1 Instituto Pirenaico de Ecología (IPE-CSIC), Avda. Montañana 1005, E-50059 Zaragoza, Spain
2 Natural Processes & Monitoring Team, Estación Biológica de Doñana (EBD-CSIC), Avda. Américo Vespucio 26, E-41092 Sevilla, Spain
3 Scuola di Scienze Agrarie, Forestali, Alimentari e Ambientali, Università degli Studi della Basilicata, Viale dell'Ateneo Lucano 10, 85100 Potenza, Italy
4 Departmento de Sistemas Físicos, Químicos y Naturales, Universidad Pablo de Olavide, Crta. Utrera Km. 1, E-41013 Sevilla, Spain
5 Instituto de Ciencias Forestales (ICIFOR, INIA-CSIC), Ctra. de la Coruña Km 7.5, E-28040 Madrid, Spain
6 ETSI Montes, Forestal y del Medio Natural, Universidad Politécnica de Madrid (UPM), C/Ramiro de Maeztu, 28040 Madrid, Spain
* Correspondence: jjcamarero@ipe.csic.es; Tel.: +34-976-369-393

**Abstract:** Assessing post-fire recovery is essential to forecast how ecosystems will respond to future warmer conditions and higher fire severity. Such assessments must consider site conditions and the post-fire recovery of trees and shrubs. We used tree-ring data and intrinsic water-use efficiency (WUEi) to quantify the post-fire responses of a tree (*Pinus pinea*) and a shrub (*Juniperus phoenicea*) in Mediterranean coastal dunes located in the Doñana protected area, SW Spain. We compared pines and junipers growing in an inter-dune slack with those growing in a nearby dune ridge. We quantified crown damage and bark char in pines impacted by a wildfire. Pines with lower crown damage after the fire showed a higher growth rate one year later. Growth decreased afterwards in the case of fast-growing pines from the slack site, whereas slow-growing dune pines showed increasing growth rates one to three years after the fire. The fire did not change the pines' WUEi, whereas the junipers located in dunes presented the highest WUEi values. Severe crown damage (damage > 60%) impairs long-term growth recovery in *P. pinea*. Open and heterogeneous landscapes can reduce the wildfire risk in the study Mediterranean area, where slack vs. dune and tree vs. shrub differences should be considered in post-fire management plans.

**Keywords:** crown scorch; dendroecology; drought; *Juniperus phoenicea*; *Pinus pinea*; post-fire response; WUEi

## 1. Introduction

Warmer and drier conditions amplify the risk of wildfire occurrence, particularly in fire-prone, climate-change hotspots such as the Mediterranean Basin [1–3]. During the last decades, wildfire frequency and severity and burnt area have increased in this region, particularly in southern Europe. This is dramatically illustrated by the late July 2022 heatwave and related wildfires affecting several countries such as Spain. Forest wildfires raise tree mortality rates and negatively impact several ecosystem services including timber production, biodiversity, water provision and carbon uptake [4]. Therefore, we need reliable assessments on the ability of trees and shrubs to recover after fires in order to adjust management measures such as post-fire salvage, planting and prescribed fires.

Comparing the post-fire recovery of different species and growth forms is essential since warmer and drier conditions and more severe and frequent fires could shift forested ecosystems to non-forested ecosystems dominated by trees and by shrubs or grasslands,

respectively [5]. In addition, woody plant species have multiple traits (e.g., reproductive precocity, thick bark, tall canopies, serotiny, etc.) and strategies (e.g., seeders vs. sprouters) to withstand fire [6,7]. For instance, *Pinus halepensis* Mill. shows an early reproductive age and produces abundant serotinous cones, which open at high temperatures, forming a large canopy seed bank, whereas *Pinus pinea* L. (hereafter pine) does not [7]. In the case of shrubs, many species show resprouting ability after fires, such as *Juniperus oxycedrus* L., whereas others of the same genus such as *Juniperus phoenicea* L. (hereafter juniper) do not [8]. Fire adaptive traits to regenerate after fires are widely studied [9], but lesser attention has been given to the potential effects of wildfire for fire-resistant (pine) and fire-evader (juniper) species coexisting in fire-prone ecosystems.

Post-fire recovery also depends on the fire regime, intensity and type (e.g., crown vs. surface fires), as well as on fuel loading [1]. For example, in sites with sandy soils where the soil water retention capacity is low, it could be expected that the access to soil water and productivity—and, therefore, the fuel amount—would be higher in slack bottoms than in ridges. This is the case of coastal dune Mediterranean ecosystems, where trees and shrubs depend on how deep the water table is and rely on different seasonal precipitation [10,11]. Different root systems and soil water availabilities determine productivity in such coastal, drought-prone habitats, as has been demonstrated in the Doñana protected area, located in SW Spain [10]. There, mixed pine-juniper stands abound, but smaller junipers are very dependent on shallow soil moisture and have presented recent dieback in dune areas [12,13]. This could be explained by the shallow root system of the juniper, which makes it vulnerable to very warm conditions when the soil is dry [14,15]. In contrast, the pine dominates valley bottoms, with more shallow and abundant soil water forming relatively closed stands which are very vulnerable to crown fires.

In June 2017, a crown wildfire severely affected pine-juniper stands in the Doñana protected area [16,17]. This study aims to quantify the post-fire changes in the radial growth and intrinsic water-use efficiency (WUEi)—a proxy of how much water is lost through transpiration during photosynthesis—of trees (pine) and shrubs (juniper). We use a retrospective approach based on tree rings and compare two contrasting topographical conditions—dune ridges vs. inter-dune valley slacks—with different access to soil water. We also explore if the post-fire recovery of pines was related to two types of conspicuous fire damage signals, namely, bark char and crown damage. We expect a better recovery of less damaged, fast-growing pines growing in the inter-dune slacks compared with slow-growing pines occupying dune ridges. However, we expect the opposite pattern in the case of junipers because dune junipers were probably less affected by the fire since they inhabit more open areas with lower productivity, a lower amount of fuel and a higher distance between tree and shrub individual crowns. We also expect a reduction in the WUEi of inter-dune pines due to a decrease in tree-to-tree competition for soil water and nutrients.

## 2. Materials and Methods

### 2.1. Study Area and Wildfire Features

On 24 June 2017, a wildfire spread to 10,334 ha [16,17], affecting the Las Peñuelas site (Huelva province, SW Spain) located inside Doñana Natural Park, which is considered one of the most valuable conservation areas in Spain due to its high plant and bird diversity, among other values (Figure 1). Almost half of the affected area was severely impacted by the fire, and 88% of the burnt area corresponded to pine-juniper woodlands growing on coastal dune systems [16]. Based on remote sensing data, the wildfire intensity was high and moderate in 47% and 43% of the total affected area, respectively [16,17].

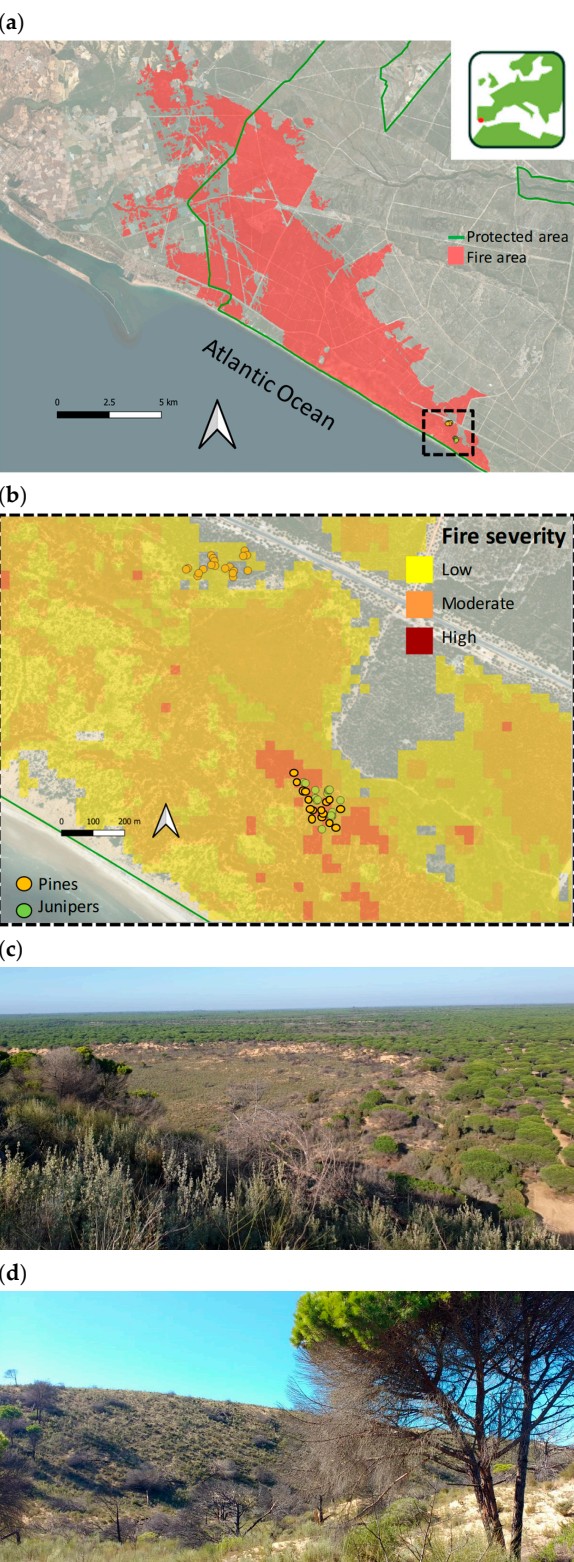

**Figure 1.** Location and landscape view of the study wildfire. (**a**) Area affected by the wildfire (red patches) in SW Spain, inside the Doñana protected area (Doñana Natural Park, green lines). (**b**) Map of fire intensity in the study area showing the location of the sampled pines (orange points) and junipers (green points), respectively; the dashed box in figure (**b**) corresponds to that shown in figure (**a**). (**c**) View of the burnt pine stand located in the inter-dune valley slack ("Corral del Muerto"). (**d**) View of a burn area in the dune sites.

In the Doñana protected area, the juniper dominates the vegetation on mobile dunes [16]. The climate is Mediterranean, with a mean annual temperature of 17.8 °C and a total annual precipitation of 540 mm [13]. The annual climate water balance is −734 mm. The understory is dominated by a diverse scrub community with species such as *Halimium halimifolium* (L.) Willk., *Cistus salvifolius* L., *Lavandula stoechas* Lam., *Salvia rosmarinus* (L.) Schleid., *Thymus mastichina* L. and *Calluna vulgaris* (L.) Hull. [18]. The soils are sandy and acid.

In the inter-dune valley slack, the pine is dominant, forming open stands with a sparse understory. This pine species is well adapted to sandy, nutrient-poor, acid soils [19,20]. The thick pine bark and lower branch self-pruning make adults of this species resistant to low-intensity fires. This could explain why this pine species lacks some post-fire resilience strategies such as reproductive precocity (early flowering and fruiting), sprouting and cone serotiny [7]. This pine produces large, heavy seeds, usually dispersed by animals, with enough reserves facilitating seedling establishment [19]. In Doñana, pines allow birds to perch, thus facilitating juniper recruitment. Planted pine stands are more productive in sites with deeper and wetter soils [16].

Nowadays, pine stands in the study area correspond to reforestations carried out in the 1940s and 1950s to fix dunes and produce timber, charcoal and pine nuts across ca. 15,000 ha, albeit the first plantations of this species started in the late 18th century [21,22].

The study area was managed as a hunting reserve since the 14th century, but the intensification of timber extraction led to a reduction in forest cover in the late 17th century, restricting the presence of pine to the southern tip of the Doñana dune system [23]. The paleobotanical evidence attests to the presence of pine species in the study area during the past 6000 years [24,25]. Lastly, fires have played an important role as a major disturbance factor in the study area, at least since the 18th century [26].

*2.2. Field Sampling*

In October–November 2020, we selected and sampled two sites affected by the 2017 wildfire. One site was located in an inter-dune valley slack ("Corral del Muerto"), and the other site ("Duna") was situated in its nearby dune ridge (Table 1, Figure 1). These sites had low- to moderate-severity fire, representing two contrasted situations for survival trees in the burned area [16,17]. We did not choose sites with a high fire severity because most trees were dead, and we could not assess post-fire growth responses.

**Table 1.** Geographical and topographical characteristics of the two study sites.

| Site Name (Type) | Latitude (N) | Longitude (W) | Elevation (m a.s.l.) | Slope (°) |
|---|---|---|---|---|
| Corral del Muerto (slack) | 37.06007 | 6.65684 | 35 | 0–5 |
| Duna (dune) | 37.05849 | 6.65761 | 70 | 20–45 |

We sampled 30 dominant pines showing different intensities of crown damage in the inter-dune valley (Figure 2) and 17 additional pines in the dune, where few surviving pines remained (Table 2). In the case of junipers, 20 individuals were selected and sampled in the dune, and 17 were selected and sampled in the inter-dune site. Since this juniper species does not resprout after fires, we selected 12 living individuals and 14 apparently dead individuals presenting no living photosynthetic tissue. The death of those individuals was further checked through tree-ring dating of the outermost, last-formed ring. In the case of living junipers, it is possible that some stems were killed by the fire, but most of the remaining stems were still alive during sampling. We observed in their cross-sections partial cambium dieback due to fire damage (Figure 2). In general, we tried to sample couples of neighboring individuals with different crown damage so as to diminish environmental variation.

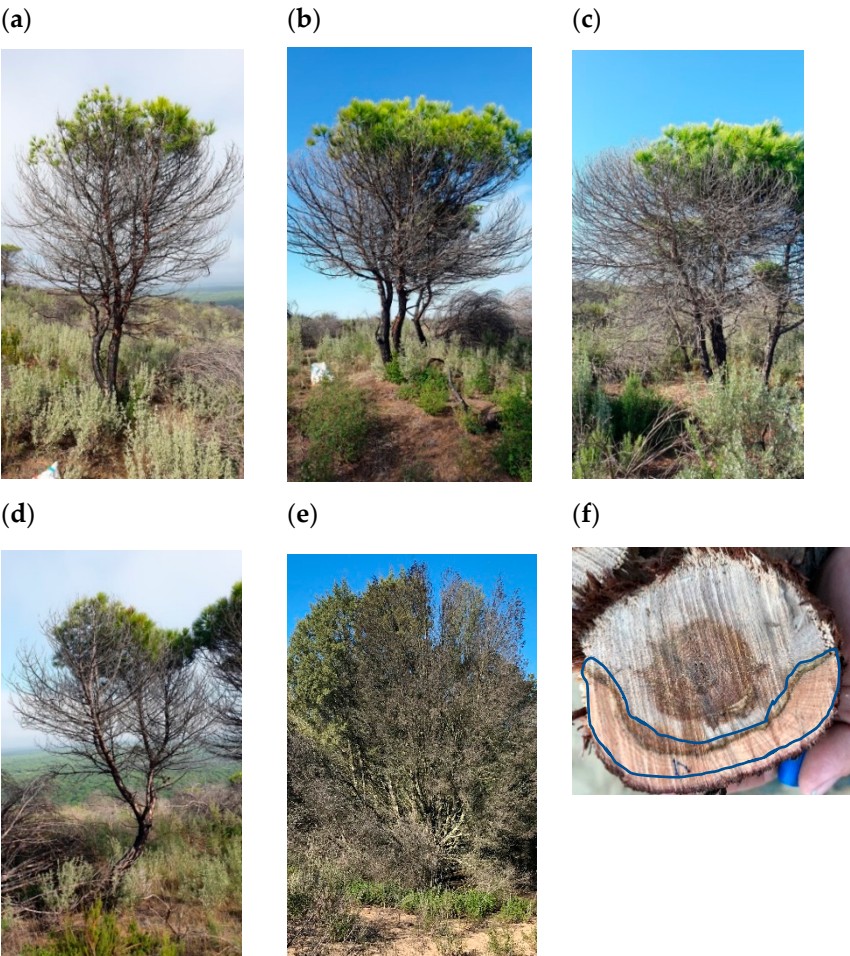

**Figure 2.** Different degrees of fire-induced crown or stem damage in the sampled pines and junipers: (**a**–**c**) pines showing moderate crown damage; (**d**) severe crown damage; (**e**) crown damage in juniper; (**f**) juniper cross-section showing partial cambium dieback (the living part of the stem is delimited by the blue line).

We measured the diameter at breast height (DBH) and the pine or juniper height using tapes or a laser rangefinder (Nikon Forestry Pro). We visually estimated the percent crown damage in pines and junipers. Post-fire tree mortality has been associated with several damage signals related to injury such as crown scorch (needles killed from the heat of a fire), overall crown damage (i.e., crown scorch and crown torch, where the needles are consumed), bark char and cambium damage [27]. Crown scorch is easily assessed in the field, and it is often used as a post-fire mortality predictor [28]. Measurements of fire-induced injury used in tree mortality models often include the percentage crown volume or the length scorched, the percentage crown volume killed, the char on bark (a proxy for cambium injury) and the bark char height [29].

Here, we followed the definition proposed by [30] and assessed total crown damage or injury, which is defined as the portion of the crown (foliage, buds, branches) killed or injured by fire, i.e., the sum of scorching, killing and consumption. In the case of pines, we also measured other injuries associated with fire, such as the surface of bark char and the bark char height, and then calculated the percentage of stem volume with bark char, assuming that pine stems have a conical shape. In other words, we calculated the relative volume of the fire scars ("catfaces"). We also annotated the exposure of the main fire scar in the stem.

We took two cores at 1.0–1.3 m from pines using a Pressler increment borer and avoided sampling near the fire scar to obtain representative samples of post-fire growth data. In the case of junipers, basal cross-sections from dead or living stems were cut using

a chainsaw. All of the sampled junipers were multi-stemmed, so taking a cross-section from a main, thick stem did not threaten the survival of living individuals. Most juniper individuals in the inter-dune site were dead.

### 2.3. Processing Wood Samples and Tree-Ring Width Data

The cores or cross-sections were air dried and carefully sanded to clearly distinguish annual rings following standard procedures in dendrochronology [31]. Then, the samples were visually cross dated using marker or characteristic rings, particularly narrow rings corresponding to dry years [32]. The outermost ring on the cross-dated samples was considered as the year in which a juniper died due to the fire. A juniper was considered to be dead if the last ring in both measured radii did not reach the sampling year (2020). Then, the ring widths were measured with a 0.001 mm resolution on the cores' or cross-sections' images obtained in a scanner (Epson Expression 12000XL) at a 2400 dpi resolution. We measured two cores per pine and two radii per juniper cross-section using the CDendro-CooRecorder software [33]. The visual cross-dating was checked using the COFECHA software (ver. 6.06P, Laboratory of Tree-Ring Research, The Univ. of Arizona, AZ, USA), which calculates shifting correlations with a mean series of each species [34].

We also calculated several statistics to characterize these series, such as the mean sensitivity (MSx) or relative difference in width between consecutive rings, the mean of correlations among all radii (rbar) and the expressed population signal (EPS), which measures how coherent and replicated a chronology is [35].

The ring-width measurements were transformed into basal area increments (BAI) by assuming a circular shape of stems as:

$$\mathrm{BAI} = \pi \, (R^2_t - R^2_{t-1}) \tag{1}$$

where $R$ is the tree radius and $t$ is the year of ring formation. The BAI was calculated using the package dplR [36] in the R statistical software [37].

### 2.4. Calculating Intrinsic Water-Use Efficiency (WUEi)

We selected ten pines and five junipers whose cross-dated ring-width series presented correlations higher than the rbar (Table 3). We considered slack (n = 5 individuals) and dune (n = 5 individuals) pines. In the case of junipers, this was only done in the dune site where several junipers survived and formed the 2020 ring (n = 5 individuals). The wood from cores (pines) or cross-sections (junipers) was separated in 3-year periods, considering the rings formed before (2015–2017) and after the fire (2018–2020) using scalpels. We used wood tissue for $\delta^{13}$C analyses, as studies comparing whole wood and cellulose show similar WUEi trends [38]. Then, wood aliquots (0.001 g) were weighed on a microbalance (AX205 Mettler Toledo, OH, USA), encapsulated into tin foil capsules and combusted to $CO_2$ using a Flash EA-1112 elemental analyzer interfaced with a Finnigan MAT Delta C isotope ratio mass spectrometer (Thermo Fisher Scientific Inc., Waltham, MA, USA). Isotope analyses were carried out at the Stable Isotope Facility (SIF, University of California, Davis, CA, USA). Stable isotope ratios were expressed as per mil deviations using the δ notation relative to Vienna Pee Dee Belemnite (VPDB). The standard deviation for repeated analyses was better than 0.1‰.

We calculated WUEi, following [39], as:

$$\mathrm{WUEi} = Ca \, [1 - (Ci/Ca)] \, 0.625 \tag{2}$$

where $Ca$ is the atmospheric $CO_2$ concentration, $Ci$ is the $CO_2$ concentration in the sub-stomatal cavity of leaves and 0.625 is the relation among the conductance of $H_2O$ compared to the conductance of $CO_2$. The $Ci$ can be calculated from $Ca$ and from changes in the $\delta^{13}$C of atmospheric $CO_2$ and wood.

The WUEi is defined as the ratio between photosynthesis rates and the stomatal conductance rate [40]. Therefore, if warm and dry conditions trigger stomatal closure and reduce stomatal conductance rates, $\delta^{13}$C and WUEi increase [38,40].

### 2.5. Statistical Analyses

To compare the BAI or WUEi data between the individuals sampled in the dune or inter-dune sites or between the pre- or post-fire periods, we used non-parametric Mann–Whitney tests. The relationships between individuals' characteristics (DBH, height, crown damage, bark char) and post-fire BAI (either for the years 2018, 2019 or 2020 or considering the mean for the 2018–2020 period) were assessed using Spearman correlation coefficients ($r_s$).

## 3. Results

### 3.1. Fire Impacts on Dune and Inter-Dune Pines and Junipers: Crown Damage and Bark Char

As expected, pines were smaller in the dune than in the inter-dune site (Table 2). In pines, crown damage and bark char also showed higher percentages in the dune than in the inter-dune site, but we found no differences between sites in the case of junipers. The mean ($\pm$ 1SE) exposure of bark char in the pine stems was 168 $\pm$ 21°, i.e., S-SE, and the mean height was 1.31 $\pm$ 0.11 m. Crown damage and the percentage of bark char were positively related ($r_s$ = 0.38, $p$ = 0.02).

**Table 2.** Characteristics of sampled pines and junipers. Values are means $\pm$ 1 SE. Different letters indicate significant ($p < 0.05$) differences between sites in each species (Mann–Whitney tests).

| Species | Site | No. Individuals | DBH (cm) | Height (m) | Crown Damage (%) | Bark Char (%) |
|---------|------|-----------------|----------|------------|------------------|---------------|
| Pine | Inter-dune | 30 | 31.9 $\pm$ 0.7b | 9.0 $\pm$ 0.3b | 46.4 $\pm$ 6.1a | 20.7 $\pm$ 2.7a |
| | Dune | 17 | 16.2 $\pm$ 0.9a | 3.4 $\pm$ 0.2a | 69.4 $\pm$ 3.7b | 33.2 $\pm$ 3.0b |
| Juniper | Inter-dune | 17 | 7.1 $\pm$ 0.3 | 2.7 $\pm$ 0.2 | 48.4 $\pm$ 4.9 | Not assessed |
| | Dune | 20 | 6.8 $\pm$ 0.4 | 2.4 $\pm$ 0.1 | 56.0 $\pm$ 6.0 | Not assessed |

In pines, crown damage was negatively related to BAI in 2018 ($r_s$ = −0.36, $p$ = 0.03; Figure 3), whereas this relationship was not observed in junipers. This could be due to the fact that many junipers (46%) found in the slack site died after the fire, i.e., their last ring was formed in 2017. The remaining 8% of junipers formed the 2018 ring (they lived 1 year after the fire), whilst the rest of the junipers (46%) survived.

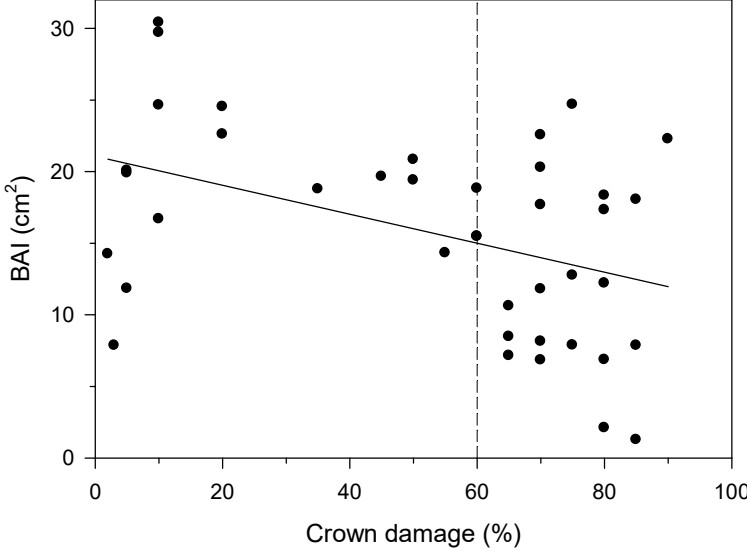

**Figure 3.** Negative relationship found between fire-induced crown damage in pines and basal area increment (BAI) one year after the fire ($r_s$ = −0.36, $p$ = 0.03). The vertical dashed line shows the 60% defoliation threshold above which growth (BAI) is significantly reduced.

In pines, the relationships between crown damage and BAI in 2019 and 2020 or the mean for the 2018–2020 period were lower, in absolute terms, than those obtained with BAI in 2018. Growth reduction was severe for crown damage above the 60% threshold. The BAI in 2018 was also positively related to the absolute volume of bark char ($r_s$ = 0.40, $p$ = 0.01), but this relationship was not significant with the percentage of bark char. There were strong, positive associations between tree size (DBH, height), estimated stem volume and radial growth (BAI). For instance, the DBH was positively related to BAI in 2018 ($r_s$ = 0.45, $p$ < 0.01) or to the mean BAI during 2018–2020 ($r_s$ = 0.53, $p$ = 0.004).

### 3.2. Fire Impacts on Growth in Dune and Slack Pines and Junipers

Considering the common and best-replicated period of 1970–2020, in pines, the mean BAI of individuals located in the inter-dune slack (6.94 ± 0.38 cm$^2$) was significantly ($p$ < 0.05) higher than that measured (2.15 ± 0.18 cm$^2$) in the individuals from the dune ridge (Table 3, Figure 4). There were noticeable drops in BAI in both series of pines corresponding to dry years such as 1981, 1995, 2005, 2012 and 2015. Increases in BAI corresponded to wet years such as 1987, 1997, 2004, 2007 and 2010–2011. After the 2017 fire, the BAI increased in both sites but then decreased only in the inter-dune pines. The 2018 BAI increase after the fire was higher, in relative terms, in inter-dune (+75%) than in dune (+45%) individuals. The mean ages (estimated at 1.3 m) of pines inhabiting inter-dune and dune sites were 58 and 43 years, respectively.

**Table 3.** Growth statistics of sampled pines and junipers calculated for the common, best replicated period of 1970–2020. The "period" column indicates the complete period. BAI values are means ± 1 SE. Abbreviations: BAI, basal area increment; MSx, mean sensitivity; rbar, mean of correlations among all radii; EPS, expressed population signal. Different letters indicate significant ($p$ < 0.05) differences between sites in each species (Mann–Whitney tests).

| Species | Site | Period | BAI (cm$^2$) | MSx | rbar | EPS |
|---|---|---|---|---|---|---|
| Pine | Inter-dune | 1954–2020 | 6.94 ± 0.38b | 0.32 | 0.57 | 0.92 |
| | Dune | 1963–2020 | 2.15 ± 0.18a | 0.36 | 0.71 | 0.97 |
| Juniper | Inter-dune | 1948–2020 | 1.13 ± 0.07b | 0.20 | 0.32 | 0.87 |
| | Dune | 1926–2020 | 0.87 ± 0.05a | 0.25 | 0.37 | 0.91 |

In junipers, the BAI pattern was similar, with dune individuals growing less than inter-dune individuals (Figure 4). However, the fire more severely impacted inter-dune junipers, where only 16% of the sampled individuals survived. The 2018 BAI decreased after the fire by −83% and −50% in the inter-dune and dune sites, respectively. The mean ages of the junipers in the inter-dune and dune sites were 61 and 67 years, respectively. As in the case of pine, BAI drops corresponded to dry years (1994, 2005), and BAI increases corresponded to wet years (1987, 2007).

Regarding growth statistics, the dune pines showed a higher mean sensitivity and coherence (rbar, EPS) among individuals than the inter-dune pines (Table 3). The inter-dune junipers showed a lower mean sensitivity and coherence than their dune counterparts. The pines showed a higher mean sensitivity and coherence than the junipers.

(**a**)

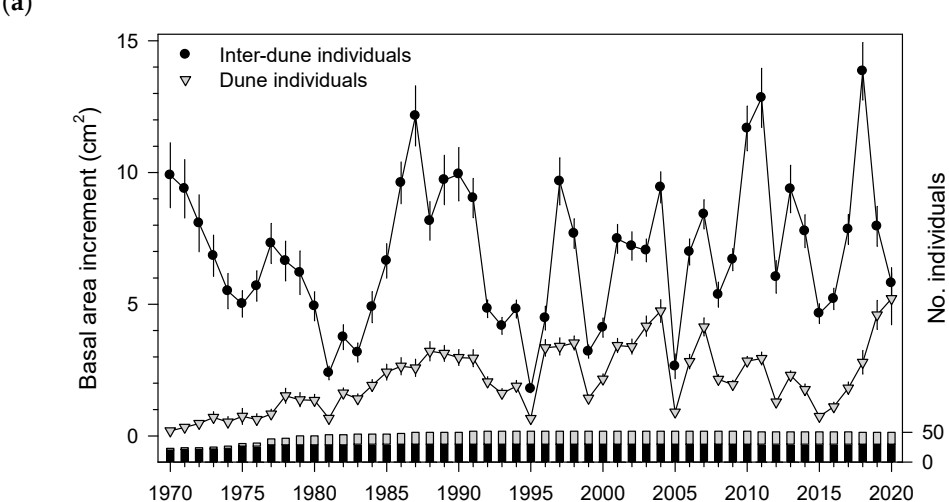

(**b**)

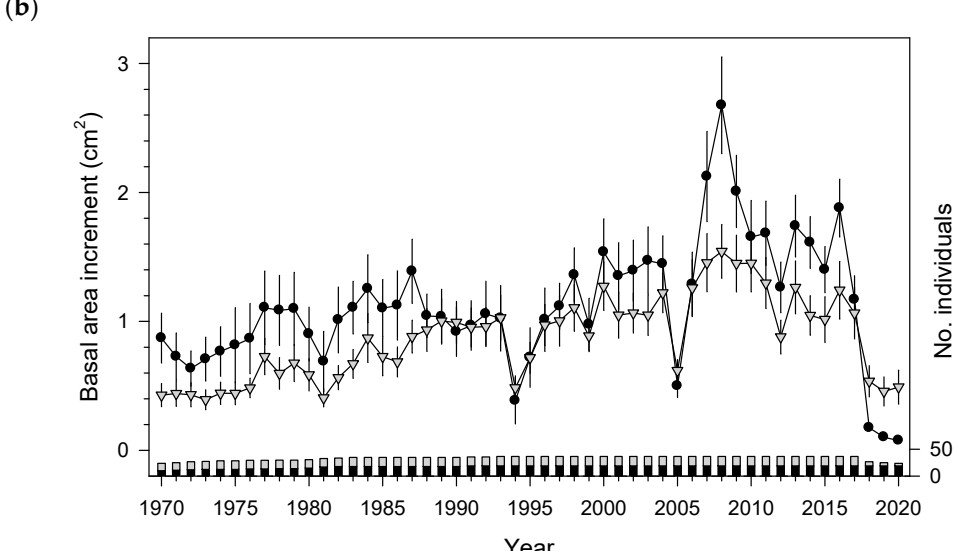

**Figure 4.** Growth patterns (basal area increment) of tree and shrub individuals sampled in or near the inter-dune valley bottom (black points) or in the dune ridge (grey points). (**a**) Pines (*Pinus pinea*); (**b**) junipers (*Juniperus phoenicea*). Basal area increment values are means ± SE. The right y axes show the number of sampled individuals (black and grey bars indicate inter-dune and dune individuals, respectively).

### 3.3. Fire Impacts on Water-Use Efficiency in Dune and Inter-Dune Pines and Junipers

In the period before the fire, the WUEi was significantly higher in junipers from the dune (118 μmol mol$^{-1}$) than it was in pines from the dune (102 μmol mol$^{-1}$) or in inter-dune pines (98 μmol mol$^{-1}$), which showed the lowest WUEi (Figure 5). After the fire, the dune junipers showed higher WUEi values (115 μmol mol$^{-1}$) than the dune (103 μmol mol$^{-1}$) or inter-dune (102 μmol mol$^{-1}$) pines, whose WUEi values did not significantly differ. There were no significant differences in WUEi before and after the fire in the case of junipers or pines.

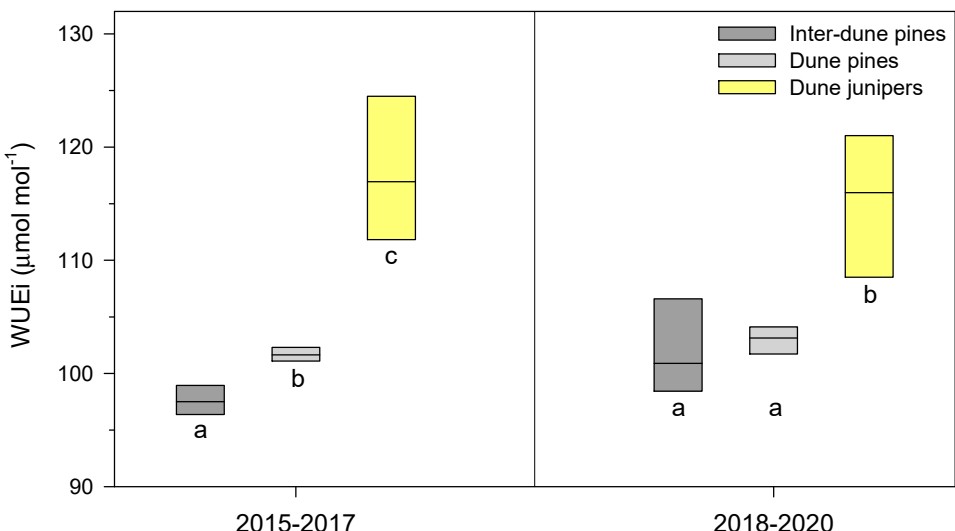

**Figure 5.** Water-use efficiency (WUEi) values calculated for pines and junipers located in the inter-dune valley bottom or in the dune ridge during the pre- (2015–2017) and post-fire (2018–2020) periods (x axis). Different letters between groups within the same period indicate significant ($p < 0.05$) differences based on Mann–Whitney tests.

## 4. Discussion

### 4.1. Fire Impacts on Pine and Juniper Growth Depend on Site Conditions

As expected, pines with lower crown damage showed higher growth rates one year after the fire. Since radial growth rates were related to tree diameter, and both were higher in the most productive slack than in the dune ridge, it is probable that the slack surviving pines will show a good growth recovery in the short term. In addition, their WUEi did not significantly increase after the fire. These patterns explain the high growth rate observed one year after the fire in the inter-dune pines but not the growth drop in 2019–2020. Such reduction in stem growth is possibly related to the dry conditions which affected the study area in 2019–2020, culminating in a drought which started in 2012 [13]. In contrast, the slow-growing dune pines showed increasing growth from 2018 to 2020 and similar WUEi values before and after the fire, suggesting a better ability to show a long-term growth recovery after the fire as compared with the slack pines.

In the case of junipers, post-fire recovery is only feasible in the dune sites, since junipers from the inter-dune site were severely damaged and many died, thus confirming our hypothesis. Nevertheless, junipers produce abundant fleshy fruits dispersed by birds, so juniper recruitment will occur among surviving pines in the slack sites. Pine recruitment is also being observed nowadays.

There have been several studies analyzing the impact of fire on the growth of pine species, particularly in the Mediterranean Basin and associated with prescribed burning [41–45]. They often reported a growth drop and a decrease in photosynthetic activity in scorched pines and also an improved water availability due to fire-induced thinning. Productivity and site water availability affected the responses since growth enhancement was not observed in the driest sites, echoing the BAI patterns of dune junipers.

In the study pine species (*P. pinea*), a high resistance to prescribed burning and surface fire was found; only the stem base was damaged, with the burned trees presenting increasing photosynthetic and stomatal conductance rates [44]. This positive response was interpreted as reduced competition for water and nutrients due to the understory burning since the pine is a drought-avoiding species with a tight stomatal control of transpiration and photosynthesis under water-limiting conditions [46]. In addition, it can uptake groundwater from deeper soil layers during the dry season [47].

The pine has a high resistance to surface fire thanks to its thick bark, providing stem insulation from heat and reducing cambium injury, but it is very vulnerable to crown

fires [48–50]. If the pine crown is scorched, as in the study case, growth reduction occurs, particularly for crown damage values above the 60% threshold. However, post-fire growth changes are also contingent on water availability (dune vs. inter-dune sites) and tree size, since the tree diameter determines the radial growth. Slack pines did not show a clear growth improvement or WUEi reduction three years after the fire, which suggests that the initial growth release in 2018 was moderated by dry conditions afterwards.

*4.2. Fire Impacts on Water-Use Efficiency: Converging Values in Dune and Inter-Dune Pines*

One of the fire impacts was leading to similar WUEi values in dune and inter-dune pines through an increase in WUEi in the case of inter-dune pines. This could be interpreted either as a reduction in stomatal conductance rates or as a reduction in both photosynthesis and stomatal conductance rates. Since these pines were severely impacted by the fire and presented high crown damage and bark char, the WUEi increase could be a stress signal due to the lower photosynthetic and transpiring leaf area. Alternatively, a reduction in tree-to-tree competition for soil moisture and nutrients, along with a reduction in WUEi and a mid-term growth improvement, could also be expected due to the fire impact on the stand. However, this was not observed and allows for the rejection of one of the posed hypotheses.

The reduced response of WUEi to post-fire open stand conditions can be caused by the 2018–2020 dry conditions already commented on. Severe soil water shortage can lead to a decrease in transpiration and carbon uptake [46]. In addition, crown damage can impair the synthesis of photosynthates and their mobilization to other functions (defensive metabolites, osmolytes) or to meristems such as the cambium [9]. We did not observe other tree damage such as dead roots or resin pith tubes associated with bark beetles in the sampled pines, which could have also explained the lack of post-fire WUEi responsiveness. In addition, the study wildfire was fast and the litter layer is shallow, which suggests that the impact of fire on pine roots was not severe enough to damage them. This would explain the predictive value of crown damage and bark char in this study case.

In terms of long-term growth recovery, the more conservative strategy of slow-growing, high-WUEi dune pines and junipers seems to be more promising. In addition, dune individuals showed higher crown damage than those sampled in the inter-dune site. This agrees with remote sensing information obtained at the stand scale, showing that the post-fire canopy cover and greenness were lower in dune than in inter-dune sites [51]. Moreover, the NDVI drop due to the fire was stronger in dunes, suggesting a higher fire severity. Nevertheless, we lacked information on the fuel amount before the fire (e.g., cover, density, basal area), which would have helped to quantify the fire severity in dune vs. slack sites.

*4.3. Management Implications and Future Research*

We used observational and statistical analyses to characterize post-fire recovery, but mechanistic models provide stronger explanations on how fire kills trees [52]. Nevertheless, our results are in line with other studies in fire-prone forests, such as giant sequoia (*Sequoiadendron giganteum* (Lindl.) J. Buchholz) stands in California (USA), where the presence of fire scars and crown damage were strong predictors of tree mortality induced by fire [27]. Overall, it has been found that crown damage is a strong predictor of tree mortality up to ten years after fires, as compared with stem char [53].

Post-fire recovery in the study area is rapid because the diverse understory is formed by several resprouter or seeder shrub species adapted to wildfires [16]. A different issue is pine forests, which form dense and homogeneous stands, leading to a high accumulation of fuel loads (e.g., needle litter) and increasing wildfire risk. Both pines and junipers do not resprout after fires and are also prone to drought-induced dieback [13]. The vulnerability of the juniper to both fires is remarkable, which also reduces its reproductive capacity [54], and dieback triggered by hot droughts, which suggests they should be better protected from wildfire damage. Therefore, a more heterogeneous and open landscape could reduce fire risk and severity and be more resilient. Pines could be managed and exploited for timber and nuts in the most productive areas with deep and wet soils whenever they are not

damaged by severe crown fires, but in less productive sites, pine stands could be thinned or replaced by diverse scrublands showing a higher resilience to fire. Nevertheless, we found a higher post-fire resilience of pines and junipers in dune sites, the least productive sites. Lastly, interactions between disturbances and new disturbance regimes should be considered since more severe droughts and wildfires could negatively impact pine-juniper dune woodlands. It has been shown that trees may be more vulnerable to delayed post-fire mortality if they are exposed to severe drought, which impairs post-fire tree recovery [55].

Future research could monitor the performance of fire-damaged stands and learn from their recovery trajectories to perform proactive management so as to reduce future wildfire risk under forecasted warmer and drier conditions, leading to more frequent fires [3,56]. It should also be tested if the replacement of some less productive, dense pine stands by scrublands or open juniper woodlands makes Doñana terrestrial ecosystems more resistant and resilient to wildfire and drought at the landscape level.

### 5. Conclusions

We found that crown damage was negatively related to post-fire growth in *P. pinea* after a severe, crown fire. Post-fire changes in growth and WUEi depended on site conditions, with slow-growing *P. pinea* and *J. phoenicea* individuals growing in the dune site showing a better growth recovery. The reduction of tree-to-tree competition in the inter-dune site, where soil water availability is higher than that in the dune site, did not translate into mid-term growth enhancement or a reduction in WUEi, probably because of crown scorch. Therefore, crown damage after severe fires can impair the long-term growth recovery of *P. pinea*.

Creating heterogeneous, more open pine-juniper woodlands in the Doñana dune systems would reduce the risk and impact of severe wildfires. Moreover, the different growing conditions provided by dune and inter-dune sites to pines and junipers should be considered when developing these management plans.

We demonstrated the resistance of pine-juniper mixed stands to moderate, surface wildfires, suggesting that low-intensity fires could be compatible with the conservation of these diverse, coastal dune ecosystems preventing crown fires. The exclusion of fires in national reserves opens the door to including prescribed fire as a conservative tool.

**Author Contributions:** Conceptualization, J.J.C. and R.D.-D.; methodology, J.J.C., M.C., C.V. and R.S.-S.; software, J.J.C. and C.V.; validation, M.C., C.V., J.M. and R.S.-S.; formal analysis, J.J.C.; investigation, J.J.C., J.M. and R.D.-D.; resources, J.J.C.; data curation, J.J.C. and C.V.; writing—original draft preparation, J.J.C.; writing—review and editing, R.D.-D., M.C., C.V., R.S.-S. and J.M.; funding acquisition, R.D.-D., J.J.C. and J.M. All authors have read and agreed to the published version of the manuscript.

**Funding:** This research was funded by the Spanish Ministry of Science, grant number CGL2015-69186-C2-1-R.

**Institutional Review Board Statement:** Not applicable.

**Informed Consent Statement:** Not applicable.

**Data Availability Statement:** Data are available on request to the corresponding author.

**Acknowledgments:** We acknowledge the support of ICTS-RBD in the sampling design and assistance, especially the Natural Processes Monitoring Team. We also thank the managers of "Espacio Natural de Doñana" for the research access provided. This study is dedicated to the memory of Nemesio Camarero López who passed away in August 2022.

**Conflicts of Interest:** The authors declare no conflict of interest.

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
