# Peer review of "Differential Post-Fire Recovery of Tree and Shrub Growth and Water-Use Efficiency in a Mediterranean Coastal Dune System"

_fire, doi:10.3390/fire5050135_

Round 1

Reviewer 1 Report

General comment:

This paper mainly studies the restoration of sand dune system along the Mediterranean coast after fire, and analyzes the response after fire from the perspective of tree ring data and internal water use efficiency. The potential effects of fire prevention and fire avoidance species coexisting in the fire prone system are mentioned, and the radial growth of trees after fire is quantified. At the same time, it also discusses whether the recovery of pine trees after fire is related to the bark scorching and crown damage. There are many innovations in the article that deserve further study. The analysis of existing problems is comprehensive. It is a worthwhile article to start with the actual wildfire fire and conduct field research. However, there are still several problems, as shown below.

- In figure 2 (f), In the illustration, it is said that the living stem of the juniper tree is represented by blue symbols and shows light brown. The color mark here is not obvious enough.

- In figure 2, There are five figures in the whole figure, four of which are about crown damage. What is more interesting is the cross-sectional situation of trees. Two cross-sectional damage figures of trees can be added here, which is more relevant to the following text.

- In section 3.1, The explanation of the growth ring formation of juniper is not clear enough.

- In figure 3, Some points in the figure deviate too much.

- In section 3.3, The difference in water use efficiency between different tree species can be clearly seen from the figure, but how does the rectangular dividing line in the figure distinguish the years?

Author Response

Comments and Suggestions for Authors

General comments:

This paper mainly studies the restoration of sand dune system along the Mediterranean coast after fire, and analyzes the response after fire from the perspective of tree ring data and internal water use efficiency. The potential effects of fire prevention and fire avoidance species coexisting in the fire prone system are mentioned, and the radial growth of trees after fire is quantified. At the same time, it also discusses whether the recovery of pine trees after fire is related to the bark scorching and crown damage. There are many innovations in the article that deserve further study. The analysis of existing problems is comprehensive. It is a worthwhile article to start with the actual wildfire fire and conduct field research. However, there are still several problems, as shown below.

  • We thank you for your positive comments on the ms. We just want to emphasize that we studied the post-fire responses but we did not deal with restorarion issues.

- In figure 2 (f), In the illustration, it is said that the living stem of the juniper tree is represented by blue symbols and shows light brown. The color mark here is not obvious enough.

  • We have delineated the living part of the stem using a blue line.

- In figure 2, There are five figures in the whole figure, four of which are about crown damage. What is more interesting is the cross-sectional situation of trees. Two cross-sectional damage figures of trees can be added here, which is more relevant to the following text.

  • Sorry, but we do not have cross-sections of the study pines.

- In section 3.1, The explanation of the growth ring formation of juniper is not clear enough.

  • We improved the description.

- In figure 3, Some points in the figure deviate too much.

  • Yes, because the growth variability among trees is high.

- In section 3.3, The difference in water use efficiency between different tree species can be clearly seen from the figure, but how does the rectangular dividing line in the figure distinguish the years?

  • We just used the vertical line to different the periods before and after the fire (indicated in the x axis).

Reviewer 2 Report

This manuscript entitled "Differential post-fire recovery of tree and shrub growth and water-use efficiency in a Mediterranean coastal dune system" studies the effects of a wildfire occured in summer 2017 on the growth and WUEi of Pinus pinea and Juniperus phoenicea in dune and inter-dune sites in Doña Natural park (SW Spain). The authors conclude that both pines and junipers of dune sites are more resilient and resistant to post-fire impacts than those of inter-dune sites. The manuscript is well-written and fits the scope of the journal FIRE. Even though the manuscript does not come up with any new mechanistic/functional insights, it provides interesting descriptive results about the post-fire recovery of two emblematic Mediterranean species. Additionally, some interesting recommendations for wildfire management of these ecosystems are given at the end of the manuscript.

Specific comments:

Line 77: I would better refer to WUEi instead of WUE throughout the whole manuscript.

Lines 95-97: Could you provide more information about the wildfire intensity, please?

Figure 1 (a): I suggest to map dune pine points even though they are located next to those of junipers.

Figure 1 (c): Could you provide more pictures of the burnt area of the dune sites, please?

Lines 134-135: Could you explain why you chose sites that suffered low to moderate fire severity?

Lines 170-172: I think it would be nice to mention here that most of juniper individuals in the inter-dune site were dead.

Figure 2: Could you provide more pictures of the bark char in pine and of the crown damage in junipers, please?

Line 233: Please, correct: "inter-dune" sites.

Figure 3: Could you illustrate here the 60% threshold, please? This is very useful information.

Lines 273-278: Could you, please, explain the drops and increases in BAI for junipers through the 1970-2020 period (i.e. as you have done for pines)?

Figure 5: I suggest showing the whiskers of the boxplots.

Line 363: Please, correct: "the lack of pos-fire WUE responsiveness".

Line 399: Please, correct: “It has been shown that …”

Line 423: Please, correct: “[…] opens the door to […]”

Author Response

Comments and Suggestions for Authors

This manuscript entitled "Differential post-fire recovery of tree and shrub growth and water-use efficiency in a Mediterranean coastal dune system" studies the effects of a wildfire occurred in summer 2017 on the growth and WUEi of Pinus pinea and Juniperus phoenicea in dune and inter-dune sites in Doña Natural park (SW Spain). The authors conclude that both pines and junipers of dune sites are more resilient and resistant to post-fire impacts than those of inter-dune sites. The manuscript is well-written and fits the scope of the journal FIRE. Even though the manuscript does not come up with any new mechanistic/functional insights, it provides interesting descriptive results about the post-fire recovery of two emblematic Mediterranean species. Additionally, some interesting recommendations for wildfire management of these ecosystems are given at the end of the manuscript.

  • We sincerely thank you for your positive comments on the ms.

Specific comments:

Line 77: I would better refer to WUEi instead of WUE throughout the whole manuscript.

  • Done, we changed it.

Lines 95-97: Could you provide more information about the wildfire intensity, please?

  • We provided information on fire intensity based on remote sensing data.

Figure 1 (a): I suggest to map dune pine points even though they are located next to those of junipers.

  • Done, we mapped them.

Figure 1 (c): Could you provide more pictures of the burnt area of the dune sites, please?

  • We added a picture showing a burnt dune area.

Lines 134-135: Could you explain why you chose sites that suffered low to moderate fire severity?

  • Because we wanted to study post-fire growth responses (in sites with high severity most trees were dead). We commented it on the revised ms.

Lines 170-172: I think it would be nice to mention here that most of juniper individuals in the inter-dune site were dead.

  • Done, we commented it.

Figure 2: Could you provide more pictures of the bark char in pine and of the crown damage in junipers, please?

  • We are sorry but we mainly took pictures of crown damage in pines. We added a picture of crown damage in juniper. We think that the images presented in figure 2 show bark char in damaged pines.

Line 233: Please, correct: "inter-dune" sites.

  • Done, we corrected them.

Figure 3: Could you illustrate here the 60% threshold, please? This is very useful information.

  • Done, thank you.

Lines 273-278: Could you, please, explain the drops and increases in BAI for junipers through the 1970-2020 period (i.e. as you have done for pines)?

  • Done, we did it.

Figure 5: I suggest showing the whiskers of the boxplots.

  • There are no outliers and the sample size is so low that whiskers can’t be plotted.

Line 363: Please, correct: "the lack of pos-fire WUE responsiveness".

Line 399: Please, correct: “It has been shown that …”

Line 423: Please, correct: “[…] opens the door to […]”

  • Done, we corrected them.